# Immunobridging Trials: An Important Tool to Protect Vulnerable and Immunocompromised Patients Against Evolving Pathogens

**DOI:** 10.3390/vaccines13010019

**Published:** 2024-12-29

**Authors:** Pedro Cruz, Jie Min Lam, Jehad Abdalla, Samira Bell, Jola Bytyci, Tal Brosh-Nissimov, John Gill, Ghady Haidar, Michael Hoerger, Yasmin Maor, Antonio Pagliuca, Francois Raffi, Ffion Samuels, Dorry Segev, Yuxin Ying, Lennard Y. W. Lee

**Affiliations:** 1Department of Medical Oncology, Portuguese Oncology Institute of Porto (IPO-Porto)/Porto Comprehensive Cancer Center Raquel Seruca (Porto. CCC Raquel Seruca), 4200-072 Porto, Portugal; 2University College London, London WC1E 6BT, UK; 3Al Rahba Hospital, HMCV+XHF, AL Shahama, United Arab Emirates; 4Division of Population Health and Genomics, University of Dundee, Dundee DD1 4HN, UK; 5University of Oxford, Oxford OX1 2JD, UK; 6Infectious Diseases Unit, Samson Assuta Ashdod University Hospital, Ashdod 7747629, Israel; 7Faculty of Health Sciences, Ben Gurion University in the Negev, Beer Sheba 8410501, Israel; 8Division of Nephrology, St. Paul’s Hospital, The University of British Columbia, Vancouver, BC V6Z 1Y6, Canada; 9University of Pittsburgh, Pittsburgh, PA 15260, USA; 10Tulane University, New Orleans, LA 70118, USA; 11Infectious Disease Unit, Edith Wolfson Medical Center, Holon 5822012, Israel; 12Faculty of Medicine, Tel Aviv University, Tel Aviv 6997801, Israel; 13Kings College London, London WC2R 2LS, UK; 14CHU Nantes, INSERM, Department of Infectious Diseases, Nantes Université, CIC 1413, 44093 Nantes, France; 15NYU Grossman School of Medicine, New York, NY 10016, USA; 16Centre for Immuno-Oncology, University of Oxford, Oxford OX3 7DQ, UK

**Keywords:** immunobridging, clinical trials, immunocompromise, vaccines, monoclonal antibodies

## Abstract

Safeguarding patients from emerging infectious diseases demands strategies that prioritise patient well-being and protection. Immunobridging is an established trial methodology which has been increasingly employed to ensure patient protection and provide clinicians with swift access to vaccines. It uses immunological markers to infer the effectiveness of a new drug through a surrogate measure of efficacy. Recently, this method has also been employed to authorise novel drugs, such as COVID-19 vaccines, and this article explores the concepts behind immunobridging trials, their advantages, issues, and significance in the context of COVID-19 and other infectious diseases. Our goal is to improve awareness among clinicians, patient groups, regulators, and health leaders of the opportunities and issues of immunobridging, so that fewer patients are left without protection from infectious diseases, particularly from major pathogens that may emerge.

## 1. Introduction

Infectious diseases continue to cause significant morbidity and mortality worldwide, despite the wide availability of medical technologies such as vaccines, treatments, or exogenous antibodies administered to prevent infections. Infectious pathogens are constantly mutating to evade human immunity, resulting in a reduction in the clinical effectiveness of these technologies with time.

Access to up-to-date effective drugs is required to ensure that vulnerable patients, including the immunocompromised, are protected from these pathogens. Patients can be immunocompromised for a myriad of reasons. These include having solid or haematological cancer, an organ transplant, kidney disease, autoimmune disease, primary and secondary immunodeficiencies, including those due to human immunodeficiency, virus infection, or immunosuppressive drugs. These individuals make up 1–5% of the population and are particularly vulnerable to infectious diseases [1]. To race against evolving pathogens, streamlined assessment and approval pathways for new medical technologies are fundamental. They empower patients to maximise their quality of life and reduce their ongoing infection risk. Such is especially pertinent for those with limited life expectancy, such as advanced cancer or end-stage kidney disease patients.

Immunobridging enables rapid, real-time access to effective technologies for evolving pathogens. It uses surrogate immunological measures to infer the effectiveness of a new drug. Although it has been used for years, such as for influenza vaccines, awareness in the general medical community can be enhanced [2]. We would argue that familiarity with this approach is urgently required, given the possibility of epidemics and pandemics increasing in the future. In these contexts, the rapid emergence of new variants and strains could render evidence from traditional phase 3 randomised trials less relevant, as evidence of clinical effectiveness against new strains quickly becomes necessary. In such cases, a phase 3b immunobridging trial is one of the most expedient, safe, and optimal strategies to evaluate the effectiveness of updated vaccines or therapeutics.

## 2. Search Strategy and Selection Criteria

References were searched in PubMed using the search terms “immunobridging” from January 2009 through February 2024. Relevant publications were also sought from the authors’ personal files. Only papers published in English were reviewed. The final reference list was based on relevance to the topic of this essay.

## 3. Immunobridging

Immunobridging is a trial methodology that infers the effectiveness of a new drug through a surrogate immunological measure of efficacy. The patient-level immune response generated by the updated technologies is assessed and compared to that of previous studies (Figure 1).

Immunobridging is important when frequent technological updates are needed to combat continuously evolving pathogens, rendering previous drugs ineffectual. It prioritises the ongoing immunological assessment of patients to deliver effective protection in a timely manner, placing patients at the centre of trial design. This strategy makes novel, targeted iterations of established drugs rapidly available while providing equivalent protection to their previous versions.

Immunobridging trials are appropriate for interventions that rely on pathogen-directed humoral immunity or passive prophylaxis with antibodies, as these outcomes can be evaluated using laboratory assays. Immunobridging consists of inferring the efficacy of a new drug if it can induce similar antibody titres to another drug previously shown to have clinical efficacy.

## 4. Applications

The potential applications of immunobridging trials are diverse, and they have been integral to the development of vaccines and therapies for various infectious diseases for several decades [3,4]. Notable examples include the pneumococcal [5,6] and yearly influenza vaccine updates. Annual influenza vaccines are made available not based on the results of large, expensive, and time-consuming phase 3 trials that need to be repeated annually, but rather on evidence of inducing antibody titres previously shown to be clinically effective [7,8,9,10]. Immunobridging has also been used to update human papillomavirus (HPV) vaccines to ensure that newer vaccines can cover more strains of HPV at the right doses, or to propose simpler schedules with fewer doses [11,12,13]. More recently, regarding COVID-19, immunobridging trials revealed many advantages. They have allowed the assessment of newer vaccines [14,15,16] and the reassessment of older vaccines [17]. Importantly, they have facilitated the rapid update of COVID-19 vaccines, such as those against variants like Omicron, and subvariants including XBB. The initial clinical trials of mRNA-1273 and BNT162b2 enrolled 30,420 and 43,548 participants, respectively [18,19], while more recent immunobridging trials of updated bivalent Omicron-containing vaccines enrolled 819 patients [20]. The lower number of participants needed is one of the advantages of immunobridging trials [21]. So is the shorter trial duration, which accelerates review and approval by regulatory authorities, increases the relevance to the current pathogen landscape, decreases investment cost, and eases clinical trial requirements (Table 1).

Beyond vaccines, immunobridging may also be highly relevant to the future of other therapies, including monoclonal antibodies. However, unlike vaccines, in which immunobridging has been utilised for decades, this approach has rarely been used for monoclonal antibodies, which are still subjected to traditional modes of review by regulatory authorities. Taking the recent history of monoclonal antibodies that were developed for SARS-CoV-2 as an example, casirivimab, imdevimab, bamlanivimab/etesevimab, sotrovimab, and tixagevimab + cilgavimab all showed great success in treatment and/or prophylaxis. However, their utility ultimately proved short-lived with the emergence of resistant variants. We suggest that this is a key area where immunobridging can contribute to adaptive trial design, timely approval, and ultimately, the accelerated deployment of efficacious monoclonal antibodies. Immunocompromised patients could draw great benefit from the advantages of immunobridging in vaccine and monoclonal antibody development, given their increased risk of infection.

## 5. Conceptual Framework

Immunobridging is a conceptually simple framework. First, a completed traditional phase 3 randomised controlled trial reaching a positive endpoint with an investigational product (vaccine, prophylactic agent, or therapeutic monoclonal antibody) is necessary. In this original trial, immunological surrogate measures of efficacy should be performed. In the case of COVID-19, these often entail the use of neutralising antibody titres. Neutralising antibody titres are an individual patient-level measure of the effectiveness of their antibodies in binding to, and neutralising, a pathogen in vitro. There is increasing evidence that neutralising antibodies are a valid correlate of protection for COVID-19, based on five different types of studies: natural history studies correlating infection-induced neutralising antibody responses with outcomes; vaccine challenge studies in humans; experimental manipulation of neutralising monoclonal antibody titre to assess mechanistic causation; efficacy trials quantifying the relationship between vaccine efficacy and neutralising antibody titre; and meta-analysis of efficacy trials correlating vaccine efficacy with neutralising antibody titre [22,23,24,25,26].

The use of neutralising antibody titres, an immunological surrogate measure, allows clinicians, regulators, and industry to correlate neutralising antibody titres with the clinical efficacy endpoint of the original clinical trial in a quantitative manner. For instance, if patients who avoided infection had an antibody titre of 100 IU/mL, that could be a surrogate marker [27,28]. If the association of meaningful clinical benefit with this correlate of protection is a suitably robust endpoint, it can act as an appropriate benchmark for future studies of an updated product, as has been demonstrated for SARS-CoV-2 vaccines [23,24,25,29]. These could be phase 3b immunobridging trials (Figure 2).

With anti-infective technologies extending beyond vaccines, both traditional (e.g., inactivated, adjuvant, and live attenuated) and novel (e.g., mRNA or adenoviral vector), into new forms of therapeutics, such as long-acting monoclonal antibodies, it is a timely moment to realise the benefits of phase 3b immunobridging trials. Such trials are innovative, as they measure the immunological function centred on the patient’s current context of previous clinical exposures (i.e., previous infection and boosters, for example). This contrasts with other outcome measures, such as hospitalisation rates, that need a long time to be ascertained and are likely affected by healthcare system capacity in addition to strain virulence, like in the first clinical trials of SARS-CoV-2 vaccines [18,19]. Rates of inpatient hospitalisation or intensive care admission are also frequently confounded by the healthcare setting and investment. Thus, immunobridging trials can provide results more rapidly, and can be applied to medical technologies beyond vaccines, such as monoclonal prophylactic antibodies. Both these aspects bring important benefits to immunocompromised patients.

Additionally, the flexibility of immunobridging trial design enables trials to be promptly deployed locally into new variant outbreak regions, and it has been proposed for highly virulent diseases like Ebola [30]. Phase 3 trials are expensive and take a long time to achieve results. In a situation of rapidly changing variants, these trials cannot be completed in a timely manner. When their results are available, the monoclonal antibodies may have already become obsolete. If immunobridging trials are accepted as the model of choice, there can be timely therapeutic updates for other pathogens with high evolutionary capacity besides influenza, like SARS-CoV-2. It would result in our patient communities having the highest level of reassurance in our capability to rapidly refine and update our global drug toolbox to protect them.

## 6. Limitations

Whilst phase 3b immunobridging trials offer a potentially powerful strategy enabling us to run trials that can respond to a rapidly evolving pathogen, no trial is perfect, and we need to be aware of potential limitations. Firstly, immunobridging is reliant on global systems to monitor pathogen evolution, and this requires diagnostic infrastructure to identify variants. Secondly, immunobridging depends on the reference neutralising antibody titres of previous trials, which in turn can have important variability. These titres differ between viral variants [31] and wane over time [32,33]. They vary between clinical endpoints, such as (severe) infection rate, seroconversion rate, duration of protection, and hospital or intensive care unit admission rate [23,34,35,36,37]. Furthermore, this approach requires an acceptance that current methodological approaches are reliant on the level of antibody, rather than affinity to the specific mutated antigen. Thirdly, not all pathogens have robust correlates of protection, and there are challenges in obtaining real infectious viruses and in interpreting pseudovirions or cultured virus assays. Fourthly, the neutralising antibody titre assays of the original trial must be similar or comparable to the assays of the secondary trial. While criteria for non-inferiority for such trials have been well established, with a lower bound of the confidence interval above the non-inferiority threshold of −10% [38,39], there is no rigorous standardised method for neutralising antibody titres measurement in vitro. The measurements depend on the assay used, sometimes with great variability, even after standardisation to international units [40,41]. It also follows that immunobridging trials are appropriate for measurements of humoral immunity only, which requires the vaccine or monoclonal antibody to be directed against a pathogen that has been well neutralised by the humoral immune response. Furthermore, current immunological assays may be unable to ascertain or differentiate vaccine-elicited and exogenously elicited antibody responses. Fifthly, the confidence in the findings of an immunobridging trial can be highest when the investigational product update is not significant and is well defined biologically. For example, a minor update to a vaccine epitope without changes to the vector, or updates to the arms of a monoclonal antibody whilst maintaining the fragment crystallisable domain, are ideal candidates for immunobridging trials. Small changes may also support similar safety profiles of the products. Sixthly, in immunobridging trials, the immunological surrogates might not predict the side effects of the new drug. For example, the various anti-amyloid beta antibodies achieve different levels of amyloid clearance and have different incidences of brain haemorrhage [42,43,44]. Thus, one must still consider the toxicities of new drugs in immunobridging trials. Finally, while immunobridging trials can quickly reduce immunocompromised patients’ infection risk, it is unlikely they will eliminate that risk entirely.

## 7. Conclusions

Immunobridging, a patient-centred trial design, offers a potentially promising, rational, and relevant strategy to provide patients with greater and more expedient protection against infectious diseases. This may include pathogens that could cause future outbreaks and pandemics, as well as infections that are endemic in our communities (e.g., respiratory syncytial virus, influenza, and hepatitis). Phase 3b immunobridging trials are an established approach with a long history of success in vaccine development. The key limitation is confidence in the use of surrogate immunological markers, and an important challenge to support is future work required to build awareness and expertise. As demonstrated by its application to influenza and COVID-19, this approach prioritises patient safety and could improve well-being. Such phase 3b trials are crucial to achieving expedited updates for new vaccines, monoclonal antibodies, or small molecule immunomodulators. They are a tool to help the clinical community maintain continual access to the advanced therapies required to protect their patients. This is particularly of relevance to immunocompromised patients, who have lower levels of protection from infectious pathogens, resulting in both physical and psychological morbidity. Our recent shared experience with the SARS-CoV-2 pandemic has re-highlighted the importance of streamlining trial delivery to update therapeutic products to protect immunocompromised and vulnerable individuals. Furthermore, it is heartening that the regulatory community has developed a high level of alignment and confidence in immunobridging. The International Coalition of Medicines Regulatory Authorities (ICMRA) members agreed that well-justified and appropriately designed immunobridging studies are an acceptable approach for authorising COVID-19 vaccines. The FDA approved a long-acting monoclonal Antibody for Pre-exposure Prevention of COVID-19 based on immunobridging data [27,45,46].

Noting this landscape, immunobridging trials can become a new standard of trial design for rapidly evolving pathogens where timely therapeutic updates are required. Moreover, it could also be positively impactful if there is a need for data bridging among different countries and regions. This may be relevant if there were a new variant emerging in one nation, and there was a requirement to infer the effectiveness of a new therapy in another nation. If delivered carefully in conjunction with the global clinical and academic community, immunobridging can be a rigorous approach to reduce patient suffering from infections, offering the promise of a healthier, more secure future where patients are empowered to live richer and safer lives.

## Figures and Tables

**Figure 1 vaccines-13-00019-f001:**
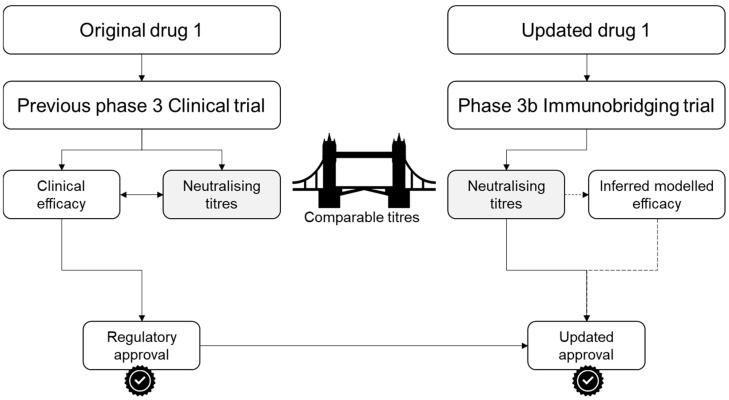
Immunobridging paradigm.

**Figure 2 vaccines-13-00019-f002:**
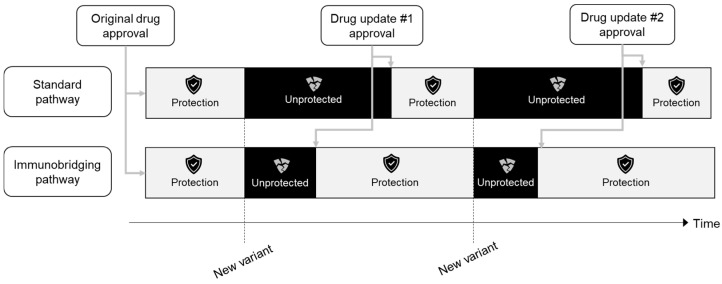
Phase 3b immunobridging trials accelerate drug update and approval, thus decreasing the timespan in which patients are unprotected against new variants.

**Table 1 vaccines-13-00019-t001:** Comparison of phase 3 randomised controlled trials and phase 3b immunobridging trials.

Parameter	Phase 3 randomised controlled trials	Phase 3b immunobridging trials
Pathogens	Against historical pathogens	Against current pathogens or novel variants or historical pathogens
Trial size	1000–10,000	100–1000
Outcome	Clinical outcomes dependent on pathogen virulence and healthcare system capability, with long follow-up (e.g., hospitalisation and intensive care admission)	Immunological profiles and outcomes dependent on patient’s immune system, such as vaccines, or on the pharmacokinetics of the administered products such as monoclonal antibodies (e.g., neutralising antibody titres)
Comparator	Usually, placebo	Previous versions of the product
Patients	Historical participants who may not have had previous infection, vaccines, or boosters	Patients who are representative of the current situation
Investment cost	Sizeable	Modest
Speed of result	1–3 years	0–1 year
Location	Often across multiple countries	Flexible. Can be spread across multiple countries or deployed locally into outbreak regions
Trial requirements at hospital trial sites	Highly intensive with ongoing clinical review and the need for patient assessment	Often less intensive, with fewer trial requirements
Therapeutic	Patients may receive a placebo or the best therapeutic that is widely used in clinical practice as the comparator	Flexible. Dependent on design, more/all patients could receive the updated therapeutic

## Data Availability

No new data were created or analyzed in this study. Data sharing is not applicable to this article.

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
