# Peer review of "Immunobridging Trials: An Important Tool to Protect Vulnerable and Immunocompromised Patients Against Evolving Pathogens"

_vaccines, 2024, doi:10.3390/vaccines13010019_

Round 1
Reviewer 1 Report
Comments and Suggestions for Authors
In the review titled “Immunobridging trials: an important tool to protect vulnerable and immunocompromised patients against evolving pathogens”, the authors intended to broaden the awareness of the immunobridging trial method and introduce this new methodology to the general public including clinicians, patient groups, regulators, and health leaders. The focus and framework of this review need some modifications before the manuscript can be accepted for publication.
1. The title and the introduction section define the premise as protecting the immunocompromised patient. However, the main content is mainly about the concept of immunobridging. It did not explain how immunobridging would affect immunocompromised individuals in the sections of application, framework or limitation. The title needs to be changed if a general introduction of immunobridging is the focus of this review.
2. The second paragraph of Introduction (lines 47-57) seems to emphasize the necessity of exploring the potential of immunobridging trial method for a rapid assessment of new vaccine efficacy. Then the third paragraph (lines 58-77) is switched back to the commonly accepted justification: to race against the rapid emergence of new variants by accelerating the approving process. The sections 2 and 3 seemingly introduce the general definition. I suggest these few sections are rearranged and refocused. For example, Figure 1 seems to go better with section 3, and there are some repetitions of ideas laid out in the third paragraph of introduction and section 3.
3. It would help to have an explanatory figure legend for each figure, or to explain the figure in the text. I do not follow what Figure 2 is trying to tell...
4. Lines 170-172: limitation of patient-reported symptoms will also affect the immunobridging method because the referenced original trial also relied on such patient-reported symptoms.
5. Line 192: starting from “Secondly,…” I have no idea what the point is from this long sentence. Shouldn’t the neutralizing antibody in immunobridging trial be generated from the new emerging variant, also wane over time and be affected by the various factors?
Comments on the Quality of English LanguageSome sentences are way too long. The structure of the whole manuscript is not entirely clear.
Author Response
Comments 1: The title and the introduction section define the premise as protecting the immunocompromised patient. However, the main content is mainly about the concept of immunobridging. It did not explain how immunobridging would affect immunocompromised individuals in the sections of application, framework or limitation. The title needs to be changed if a general introduction of immunobridging is the focus of this review.
Response 1: Thank you for pointing this out. We agree with this comment. Therefore, we have developed how immunobridging would affect immunocompromised individuals in the sections of application (lines 152-154), framework (line 198) and limitation (lines 248-258).
Comments 2: The second paragraph of Introduction (lines 47-57) seems to emphasize the necessity of exploring the potential of immunobridging trial method for a rapid assessment of new vaccine efficacy. Then the third paragraph (lines 58-77) is switched back to the commonly accepted justification: to race against the rapid emergence of new variants by accelerating the approving process. The sections 2 and 3 seemingly introduce the general definition. I suggest these few sections are rearranged and refocused. For example, Figure 1 seems to go better with section 3, and there are some repetitions of ideas laid out in the third paragraph of introduction and section 3.
Response 2: Thank you for pointing this out. We agree with this comment. Therefore, we have rearranged and refocused these sections, removing the repetitions of ideas in the third paragraph of the Introduction and section 3 as recommended. Thus:
- The second paragraph of Introduction was edited (lines 54-59) to better integrate the ideas of rapid assessment and approval of new vaccines to race against the rapid emergence of new variants in the second paragraph of Introduction.
- The third paragraph of the Introduction was edited (lines 60-71) and now contextualises the concept of Immunobridging, before it is defined in detail in sections 2 and 3.
- The section 3 was edited (lines 91-109) to present the general definition.
As suggested, Figure 1 was placed in section 3.
Comments 3: It would help to have an explanatory figure legend for each figure, or to explain the figure in the text. I do not follow what Figure 2 is trying to tell.
Response 3: Thank you for pointing this out. We agree with this comment. Therefore, we have edited the legend for the figures: Figure 1 legend (line 96) is now “Immunobridging paradigm.”, and Figure 2 legend (lines 179-181) is now “Phase 3b immunobridging trials accelerate and neutralising titres as a tool to drug update drugs and approvals, thus decreasing the timespan in which patients are unprotected against new variants.”
Comments 4: Lines 170-172: limitation of patient-reported symptoms will also affect the immunobridging method because the referenced original trial also relied on such patient-reported symptoms.
Response 4: Thank you for pointing this out. We agree with this comment. Therefore, we have edited lines 189 and 194-196 in order not to suggest that immunobridging trials are unaffected by the shortcomings of patient-reported symptoms.
Comments 5: Line 192: starting from “Secondly,…” I have no idea what the point is from this long sentence. Shouldn’t the neutralizing antibody in immunobridging trial be generated from the new emerging variant, also wane over time and be affected by the various factors?
Response 5: Thank you for pointing this out. We agree with this comment. Therefore, we have edited lines 215-222 to better clarify this idea and to break into different sentences. It is now “Secondly, immunobridging depends on the reference neutralising antibody titres of previous trials, which in turn can have important variability. These titres differ between viral variants [31] and wane over time [32,33]. They vary between clinical endpoints, such as (severe) infection rate, seroconversion rate, duration of protection, hospital or intensive care unit admission rate [23,34–37]”.
Comments on the Quality of English Language: Some sentences are way too long. The structure of the whole manuscript is not entirely clear.
Response 5: Thank you for pointing this out. We agree with this comment. Therefore, we have restructured sections 1 to 3 (as described in Response 2) to make the structure more clear, as suggested.
Furthermore, the writing was reviewed by a colleague of ours with Certificate of Proficiency in English, and the long sentences were reorganized into shorter sentences (lines 48-52, 54-71, 74-79, 86, 98-109, 111, 120, 126-128, 130-131, 134-138, 143-144,149, 152-154, 156-157, 159, 179-181, 182, 185-191,200-201, 204, 215-223, 226-230, 263, 266-268. 273-274.
Reviewer 2 Report
Comments and Suggestions for Authors
This article explores the concepts behind immunobinding trials, which are a novel methodology to examine the vaccine efficacy in patients using immunological markers. This article focused on the comparison of the traditional neutralizing titres and immunobridging pathway. I believe that article is well-designed to explain the immunobridging and has valuable and promising meaning for the immunobridging. In addition, the authors discussed the limitations of this technique such the the antibodies’ titer to the viral variants, the international standards, and antibody limitations. I believe that it meets the journal requirements to be published.
Author Response
Comments 1: This article explores the concepts behind immunobinding trials, which are a novel methodology to examine the vaccine efficacy in patients using immunological markers. This article focused on the comparison of the traditional neutralizing titres and immunobridging pathway. I believe that article is well-designed to explain the immunobridging and has valuable and promising meaning for the immunobridging. In addition, the authors discussed the limitations of this technique such the the antibodies’ titer to the viral variants, the international standards, and antibody limitations. I believe that it meets the journal requirements to be published.
Response 1: Thank you very much for your review and comment. We agree with this comment. Nevertheless, we have addressed the comments of the other reviewers.
Reviewer 3 Report
Comments and Suggestions for Authors
The purpose of this paper is to raise awareness among physicians, patient groups, regulators, and healthcare managers about the opportunities and challenges of immunobridging approaches. The paper includes a short list of key benefits and limitations of immunobridging research. Meanwhile, I have a few comments on the content of this paper:
(1) The authors should take into account that the immunobridging methodology is currently widely used in the testing of new vaccines and immunomodulatory drugs, and many articles and methodological recommendations have already been published on this topic. Therefore, it would be advisable for the authors to address not only the general characteristics of immunobridging, but also the specific problems of this methodology, e.g. in relation to COVID-19, in more detail, given the urgency of this issue. In particular, ICMRA organized a workshop on June 24, 2021 to review the development of COVID-19 vaccines. At the same time, ICMRA focused on immunobridging, the development and use of controlled trials and protective correlates. The members of the Access Consortium agree that robust and well-planned immunobridging studies are an acceptable approach to the licensure of COVID-19 vaccines. Since then, many studies have been conducted to test new COVID-19 vaccines using immunobridging methods (e.g., PMID: 35632411, PMID: 39318203), including 3-phase immunobridging therapy and prevention studies (e.g., PMID: 38878794, PMID: 36075233). At the same time, the authors provided a very limited list of such publications in their study. A more detailed analysis of this problem may be useful for the readers of Vaccine. In this case, the article presented by the authors would be more specific and would show more signs of novelty.
(2) Section 6. Limitations. In general, the authors identified the major limitations of immunobridging, including determining the relationship between antibody levels and protection against COVID-19. The authors correctly noted that the level of protection is not strictly correlated with the neutralization titer, and thus there is no strict threshold below which individuals are not protected or above which protection is achieved. In addition, the apparent discrepancies between studies present a challenge to the use of protection curves in public health decision making. It should also be noted that a significant limitation to harmonizing protection thresholds is the lack of a rigorous standardized analysis for measuring neutralization titers in vitro. Although an international standard has been established, reported titers appear to depend on the analysis used, as would be expected due to differences in cells, virus, and measurement results. Even after standardization of measurements from different assays to international units, standardized neutralization titers may still vary by tens of times between assays. Thus, the authors could expand the Limitations section, as many limitations are not always fully addressed when using immunobridging methods.
(3) The references do not fully meet the MDPI style criteria.
Author Response
Comments 1: The authors should take into account that the immunobridging methodology is currently widely used in the testing of new vaccines and immunomodulatory drugs, and many articles and methodological recommendations have already been published on this topic. Therefore, it would be advisable for the authors to address not only the general characteristics of immunobridging, but also the specific problems of this methodology, e.g. in relation to COVID-19, in more detail, given the urgency of this issue. In particular, ICMRA organized a workshop on June 24, 2021 to review the development of COVID-19 vaccines. At the same time, ICMRA focused on immunobridging, the development and use of controlled trials and protective correlates. The members of the Access Consortium agree that robust and well-planned immunobridging studies are an acceptable approach to the licensure of COVID-19 vaccines. Since then, many studies have been conducted to test new COVID-19 vaccines using immunobridging methods (e.g., PMID: 35632411, PMID: 39318203), including 3-phase immunobridging therapy and prevention studies (e.g., PMID: 38878794, PMID: 36075233). At the same time, the authors provided a very limited list of such publications in their study. A more detailed analysis of this problem may be useful for the readers of Vaccine. In this case, the article presented by the authors would be more specific and would show more signs of novelty.
Response 1: Thank you for pointing this out. We agree with this comment. Therefore, we have added the examples of those studies (lines 126-128) and their references.
Comments 2: Section 6. Limitations. In general, the authors identified the major limitations of immunobridging, including determining the relationship between antibody levels and protection against COVID-19. The authors correctly noted that the level of protection is not strictly correlated with the neutralization titer, and thus there is no strict threshold below which individuals are not protected or above which protection is achieved. In addition, the apparent discrepancies between studies present a challenge to the use of protection curves in public health decision making. It should also be noted that a significant limitation to harmonizing protection thresholds is the lack of a rigorous standardized analysis for measuring neutralization titers in vitro. Although an international standard has been established, reported titers appear to depend on the analysis used, as would be expected due to differences in cells, virus, and measurement results. Even after standardization of measurements from different assays to international units, standardized neutralization titers may still vary by tens of times between assays. Thus, the authors could expand the Limitations section, as many limitations are not always fully addressed when using immunobridging methods.
Response 2: Thank you for pointing this out. We agree with this comment. Therefore, we have expanded the Limitations section as suggested (lines 232-234) and added the references.
Comments 3: The references do not fully meet the MDPI style criteria.
Response 3: Thank you for pointing this out. We agree with this comment. Therefore, we have corrected the references style.
Reviewer 4 Report
Comments and Suggestions for Authors
In the opinion piece "Immunobridging trials: an important tool to protect vulnerable and immunocompromised patients against evolving pathogens”, Lee and colleagues suggest that the use of immunobridging protocols could be and should be a useful tool to rapidly approve vaccines and prophylactic antibody treatments. Using immune markers (such as antibody titers, neutralization titers, etc etc), the authors argue that smaller/faster phase 3 trials would be sufficient for quantifying the efficacy of a treatment. The authors make a valid argument and use the Flu vaccines and COVID19 vaccine/antibodies as examples of successes.
However, to fully grasp the story behind vaccine/antibody trials, the authors should include more data from other trials, where immunological targets were achieved but use of the vaccine/antibody has significant negative effects. One negative to this approach is the anti-Amyloid beta antibodies, which are effective at clearing the antigen target, but result in significant side-effects including brain hemorrhage. Including information like this phase 3 trial will balance the opinion, as there are both benefits and consequences to immunobridging.
minor point: typo in line 237 "Coaltiion"
Author Response
Comments 1: In the opinion piece "Immunobridging trials: an important tool to protect vulnerable and immunocompromised patients against evolving pathogens”, Lee and colleagues suggest that the use of immunobridging protocols could be and should be a useful tool to rapidly approve vaccines and prophylactic antibody treatments. Using immune markers (such as antibody titers, neutralization titers, etc etc), the authors argue that smaller/faster phase 3 trials would be sufficient for quantifying the efficacy of a treatment. The authors make a valid argument and use the Flu vaccines and COVID19 vaccine/antibodies as examples of successes.
However, to fully grasp the story behind vaccine/antibody trials, the authors should include more data from other trials, where immunological targets were achieved but use of the vaccine/antibody has significant negative effects. One negative to this approach is the anti-Amyloid beta antibodies, which are effective at clearing the antigen target, but result in significant side-effects including brain hemorrhage. Including information like this phase 3 trial will balance the opinion, as there are both benefits and consequences to immunobridging.
minor point: typo in line 237 "Coaltiion”
Response 1: Thank you very much for your review and comment. We agree with this comment. Therefore, we have included this information and these clinical trials in the section Limitations (lines 245-250). We agree that the opinion became more balanced. The typo was corrected.
Round 2
Reviewer 1 Report
Comments and Suggestions for Authors
The authors answered all my questions.
Reviewer 3 Report
Comments and Suggestions for Authors
The authors have mostly taken the comments into account and made the necessary changes to the article, but the references need to be finalized. Below is an example of the MDPI style:
Portugal, R.; Coelho, J.; Höper, D.; Little, N.S.; Smithson, C.; Upton, C.; Martins, C.; Leitão, A.; Keil, G.M. Related strains of African swine fever virus with different virulence: Genome comparison and analysis. J. Gen. Virol. 2015, 96, 408–419.